# Resilience Improvement and Risk Management of Multimodal Transport Logistics in the Post–COVID-19 Era: The Case of TIR-Based Sea–Road Multimodal Transport Logistics

**Riqing Liao** [1,*] , **Wei Liu** [2] **and Yuandao Yuan** [3]

1   School of Business Administration and Customs Affairs, Shanghai Customs College, Shanghai 201204, China
2   College of Transport & Communications, Shanghai Maritime University, Shanghai 201306, China
3   Qingdao Customs District, China Customs, Qingdao 266001, China
*   Correspondence: liaoriqing@shcc.edu.cn

**Abstract:** The COVID-19 pandemic has severely impacted international economics and trade, including cargo transportation. As a result, enhancing the resilience of transport and logistics in the post–COVID-19 era has become a general trend. Multimodal transport, with its advantages of speed, large volume and multiple modes, has increasingly gained attention from countries worldwide. However, multimodal transport logistics is a complex and systematic process. Its smooth flow depends not only on the transport itself, but also on the efficient supervision of customs and other government departments at ports. This study employs the theory and method of a super-network to establish a model of multimodal transport logistics, which includes TIR-based sea–road multimodal transport and customs supervision relationships. Structural and resilience-related characteristics of the super-network are analyzed, and performance parameters of the super-network are proposed. A simulation analysis is conducted, and based on the results, countermeasures to improve the resilience and promote risk management of multimodal transport logistics in the post–COVID-19 era are suggested. The findings of this study provide an exploration of more effective ways to ensure the smoothness of multimodal transport logistics and improve system resilience. The study concludes with theoretical and managerial implications.

**Keywords:** intermodal transportation; TIR; super-network; robustness; risk management

## 1. Introduction

The global COVID-19 pandemic has severely impacted international economic trade and cargo transport since its outbreak at the end of 2019. The pandemic has highlighted the importance of considering various factors, such as political environment, economic conditions, pandemic control policies, and port lockdowns, in the management of international cargo transportation. As the transportation system is complex and consists of numerous links, the failure of any link can lead to disruption of the entire transportation system. In the post–COVID-19 era, promoting the recovery of international multimodal transport can be a feasible solution to mitigate the risk of logistics disruption [1,2]. It is crucial to conduct research on the resilience and risk management of multimodal transport logistics in this area [3–5].

Resilience is the ability of multimodal transport logistics to absorb potential disruption, prevent and control risks, maintain a certain level of transport and supply, and reduce disaster losses when impacted [6]. The study of the resilience of multimodal transport logistics aims to enable the transportation system to make rational and effective use of external resources to quickly recover and reach the initial operation level after being impacted by various risks. Viewing multimodal transport logistics from the perspective of resilience and risk management will help to improve the flexibility and adaptability of its structure and function. Moreover, it will enable relevant enterprises and authorities to take selective and targeted measures to reduce the damage caused by external risks.

Multimodal transportation is a highly efficient method that plays a crucial role in boosting transportation efficiency and facilitating international trade [7,8]. The integration of land and maritime transportation networks is vital for enhancing the resilience of multimodal transport logistics during the pandemic and sustaining China's position as a global hub for manufacturing, trade and logistics in the post–COVID-19 era [9]. For example, many goods from Japan and South Korea reach Chinese ports by sea and are then transported to Central Europe and Central Asia via the China Railway Express. However, the saturation of China–Europe trains has made it necessary to explore other multimodal transport options. TIR-based multimodal logistics is an excellent alternative.

TIR (Transports Internationaux Routiers; the International Road Transport Union (IRU) is authorized by the United Nations to conduct global operations and coordinate management) is currently the only global transport system operating under the coordination of the TIR Convention (the Customs Convention on the International Transport of Goods), which aims to enhance the transparency of international goods transport, reduce costs and shorten delivery times [10,11]. Its application scope has expanded from road transport to multimodal transport [12]. Authorized TIR carriers holding TIR carnets are permitted to load goods into their vehicles, which are authorized by the Ministry of Transport, in the inland port within the territory of a TIR Convention member country. Goods that have undergone customs inspections and are sealed at the port are released immediately upon reaching their destination, eliminating the need for further inspections. The national guarantee agency covers tariff losses due to damage, loss, or other factors.

Unlike the general multimodal transport system, TIR is a well-established operational system that is interdependent with customs, TIR certification bodies, the Ministry of Transport, authorized TIR carriers, guarantee agencies and industry associations [12]. Therefore, there is no need to establish temporary import and export declarations as is the case with general multimodal transport. TIR has greatly simplified customs procedures, shortened customs clearance times and has saved up to 80% of transportation time and 38% of transportation costs (according to the IRU; https://www.iru.org/what-we-do/facilitating-trade-and-transit/tir, accessed on 27 March 2023). In recent years, the number of countries acceding to the TIR Convention has gradually increased, and on 26 July 2016, China officially acceded to the TIR Convention, becoming the 70th party to the Convention. On 5 January 2017, the TIR Convention entered into force in China. On 18 May 2018, China's TIR was first launched at Dalian, a multimodal transport port. On 15 May 2019, the General Administration of Customs announced the full implementation of the TIR Convention throughout China.

The early deployment of TIR provided a guarantee for China's multimodal transportation logistics during the pandemic. However, changes in pandemic prevention policies and port lockdowns have affected TIR's operation, TIR is even suspended in some ports. In the post–COVID-19 era, the General Administration of Customs of the People's Republic of China announced that the TIR ports were fully reopened, and balancing efficient operations with effective risk management under increasing cargo transport and limited resources became a priority for customs and other port management agencies. Several questions remain: what is the difference between TIR-based multimodal transport logistics and traditional multimodal transport logistics? Are there any risks in practical multimodal transport logistics operation? How can customs, as a regulatory agency, promote the efficient operation of multimodal transportation logistics under limited resources? This study conducted research in this area, and the key research questions are raised as follows: (i) Is it possible to establish a super-network to express the TIR-based sea–road multimodal transport logistics relationships, which are different from those of ordinary intermodal transport? (ii) What are the characteristics of this super-network, and how can they be expressed using indicators? Do these indicators vary when the super-network encounters risky attacks? (iii) In TIR-based multimodal transport, is it possible for regulatory agencies to adopt some risk management methods to avoid risks and improve the resilience of the logistics network while operating efficiently? The main contributions of this study are: (i) exploring the

expression of TIR-based sea–road multimodal transport logistics using a super-network, analyzing its characteristics, and proposing four parameters that represent its performance; (ii) based on empirical analysis, establishing a 3D model of the super-network, sorting out the risks that the super-network may encounter, and simulating the changes in the performance of the super-network under risk attacks. This article also discusses the policies and actions that can be adopted by customs, as the regulatory government agency, to ensure and improve the resilience of the multimodal transport logistics system.

The remaining parts of this study are structured as follows: Section 2 conducts a literature review; Section 3 introduces the research method and model; Section 4 conducts empirical and simulation analysis; Section 5 presents the simulation conclusions; Section 6 discusses the main findings and practical policy suggestions of this paper; Section 7 makes an overall conclusion and discusses limitations and future research suggestions.

## 2. Literature Review

The research of super-network is based on the research of graph theory and network science [13], as well as complex systems and complex networks [14–16]. Researchers such as Nagurney (2002) gave the definition of a super-network, that is, a network that "originates from and is higher than the existing network" is a super-network [17]. On the basis of this definition, many studies have been carried out. The study of super-network is a "network of networks", that is, the study of the interaction between networks of different properties. The super-network model can be used as a research tool to describe and solve the interactions and influences between networks [18]. The theory and method of super-networks can be used in multi-level decision-making research [19].

Multimodal transport has been included in the construction of comprehensive transport systems by more and more countries because of its large transport volume, convenient organization and cost savings. Scholars have studied issues such as the combination of different modes of transport, the impact of infrastructure on multimodal transport, and the choice of routes for multimodal transport network. Soyres et al. studied the infrastructure of public transport in combination with China's "the Belt and Road" initiative [9]. Wang et al. predicted the impact of railway and road infrastructure in multimodal transport on economic growth [20]. Asch et al. believed that the airport was the strategic success factor of multimodal transport logistics [21]. Nitidetch et al. conducted multimodal transport network path selection based on a fuzzy risk assessment model and data envelopment analysis [22]. Li et al. studied the expansion location-routing problem of mixed-spoke multimodal transport network hubs and took carbon emissions as the prerequisite for the study [23]. Chen et al. established a three-level multi-modal transportation network to study cross-regional emergency resource distribution under demand and route reliability [24]. Selection of multi-modal transportation system in the function of regional development has also attracted research attention [25].

In addition, as the only cross-border freight customs clearance system in the world, TIR transport can greatly simplify customs clearance procedures and improve customs clearance efficiency. After extending its application to multimodal transport, it has achieved good results in practice and has attracted increasing attention [10–12].

However, the pandemic has had an impact on international transportation. Liu et al. studied the optimization of the reliable path of multimodal transportation of emergency supplies under double uncertainty [26], and Lau et al. also focused on the emergency logistics in China in the era of the pandemic [1]. Guo et al. studied cascading failure modes and attack strategies of multimodal transport network [26]. Uddin et al. studied assignment of freight traffic in a large-scale intermodal network under uncertainty [27]. Scholars such as Peng et al. and Guo et al.studied the problem of supply chain resilience in the pandemic [28,29].

During the pandemic, improving the resilience of multimodal transport networks [30], risk reduction and resilience in railroad transport for sustainable development [31], supply chain resilience [6,32,33], supply chain resilience in the oil and gas industries [34], and the resilience of multimodal urban transport networks [35] have become important research areas.

Transportation-related risk management research mainly includes supply chain logistics related risks [36–39], risk assessment of cross-border transport infrastructure projects [40], safety risk assessment based on multimodal transport networks [41], assessing hazmat risk at nodes of transport networks [42], risk propagation method research of multimodal transport networks under uncertainty [43], risk perception in transport [44], cruise ship safety management in Asian regions [45], and cascading failures and attack strategies of multimodal transport networks.

Existing research provides positive and beneficial explorations in the application of complex networks and super-network methods in transportation, the improvement of multimodal transport and its resilience during pandemics, risk management, and the role of TIR. However, an overview of existing studies shows that super-network methods are not widely used in multimodal transport. Most studies on multimodal transport focus on multimodal transport infrastructure, site selection, routes, etc., lacking practical contact with TIR, an advanced mode of transportation, research on the overall performance combination of the system, and in-depth analysis of the overall resilience improvement and risk management of multimodal transport logistics in the post–COVID-19 era.

## 3. Methods and Models

### 3.1. Characteristics of Super-Network Model of TIR-Based Multimodal Transport Logistics

In traditional multimodal transportation involving road transportation, goods are subject to inspection by customs in each country, and each customs pass requires unpacking, inspection, and sealing. If there is long queue of trucks at a border crossing, it will take several days to pass customs. In fact, if the goods are not inspected or opened en route, the speed of road transport can be quite fast. Therefore, unlike traditional multimodal transport, TIR-based multimodal transport does not require customs inspection in transit countries; efficient operation is dependent on the management of customs and other agencies and the cooperation of enterprises.

TIR-based multimodal transport logistics encompasses a diverse range of goods and enterprises. The complicated business connections among customs, goods and enterprises in the transport process form a mesh topology with a complex interaction; the TIR transport network forms the physical basis of multimodal transport logistics. If each functional unit in transportation is regarded as a network node, and the links represent commercial ties that bind them together, the constituted abstract network structure is similar to a complex network.

There are three typical characteristics of TIR-based multimodal transport logistics. First, node heterogeneity: there are customs, goods carriers, and other multimodal transport logistics components as well as goods in the network; the nodes are heterogeneous, and their total amount is large. Second, systematicness of the network: while each node in the network can make independent decisions, it relies on interactions, competitions and cooperation with other nodes for information and income, leading to a system with symbiotic confrontation and cooperative evolution. The characteristics of the nodes, such as the transport routes operated by enterprises, are constantly changing, further driving the overall dynamic changes in the system. Third, variability of the network scale: changes in transport routes, types of trucks, new goods and packaging methods lead to changes in the number of nodes. For example, the introduction of new goods into or the departure of goods from a multimodal transport system due to changes in transport methods can increase the number of nodes and connections between them, resulting in the scaling up of the network. Given these unique characteristics, it is difficult to fully express the network structure and relationships using the complex network method. New methods should be explored to capture the complexity and dynamism of the TIR-based multimodal transport logistics network.

Therefore, this paper establishes a super-network that is "above and beyond" the complex network and may be formed by it to visualize a TIR-based multimodal transport logistics network and clarify its relationship. Currently, system science, variational inequal-

ity, and hypergraphs are the main theoretical foundations for super-network research. This paper adopts the third method to examine the internal relationship and system performance of the TIR transport super-network at both a local and overall level after its establishment.

*3.2. Construction of Super-Network Model of TIR-Based Multimodal Transport*

Firstly, three-layer sub-networks are established based on node heterogeneity: (1) TIR transport supervisor sub-network, which is recorded as the $M - M$ layer sub-network. Its nodes are TIR transport supervisors, also known as customs, and the links between nodes represent their connection relationship. For example, customs from different countries jointly supervise a TIR route through the TIR–EPD system (TIR electronic pre declaration system); (2) cargo sub-network, which is recorded as the $G - G$ layer sub-network. Its nodes are different types of goods, and the links between nodes represent the connection relationships between them, such as transportation at the same time; (3) TIR carrier network, which is recorded as the $C - C$ layer sub-network. Its nodes are TIR carriers, which are enterprises transporting TIR goods. The links between nodes represent the connection relationships between carriers, such as transporting the same goods or engaging in business cooperation. All three layers of sub-networks are all complex networks.

Secondly, the mapping relationship between layers is established: (1) the mapping between nodes in the $M - M$ layer sub-network and $C - C$ layer sub-network refers to which TIR carriers enter the multimodal transport; (2) the mapping between nodes in the $M - M$ layer sub-network and $G - G$ layer sub-network refers to which goods become TIR goods; (3) the mapping between nodes in the $G - G$ layer sub-network and $C - C$ layer sub-network refers to which TIR carriers carry out transport of TIR goods.

Thirdly, a super-network is constructed by adding the mapping relationship between layers on the basis of a three-layer sub-network. The resulting network is referred to as the TIR-based Multimodal Transport Logistics Super-Network in this paper (hereinafter referred to as $TMTLSN$):

$$
\begin{aligned}
TMTLSN &= f(G_M, G_G, G_C) \\
&= G_M + G_G + G_C + E_{M-G} + E_{G-C} + E_{M-C} \\
&= f(M, G, C, E_{M-M}, E_{G-G}, E_{C-C}, E_{M-G}, E_{G-C}, E_{M-C})
\end{aligned}
$$

In the above formula, $G_M = (M, E_{M-M})$, $M$ is the set of TIR transport supervising nodes, $E_{M-M}$ is the set of links between them; $G_G = (G, E_{G-G})$, $G$ is the set of goods, $E_{G-G}$ is the set of links between them; $G_C = (C, E_{C-C})$, $C$ is the set of goods carriers, $E_{C-C}$ is the set of links between them; $E_{M-G} = \{(M_i, G_j) | \varphi(M_i, G_j) = 1\}$ is the set of links between $M$ set nodes and $G$ set nodes; $E_{G-C} = \{(G_i, C_j) | \phi(G_i, C_j) = 1\}$ is the set of links between $G$ set nodes and $C$ set nodes; $E_{M-C} = \{(M_i, C_j) | \gamma(M_i, C_j) = 1\}$ is the set of links between $M$ set nodes and $C$ set nodes. For example, all passing countries' customs in the TIR transport route can supervise both the goods and carriers. If a carrier is registered in another customs jurisdiction, the customs in the place of registration and the customs in the transport route will form multiple supervisions, that is, there may be multiple links between $M$ and $C$ sets. As a result of TIR operation association, various links between inter-layer and intra-layer nodes can also be generated, and various items can be transported by various TIR carriers. Therefore, to simplify the expression, all business associations between two nodes are merged into a single link, regardless of the number. Nodes with larger "degree" are not highlighted in the diagram. The resulting network structure is depicted in Figure 1.

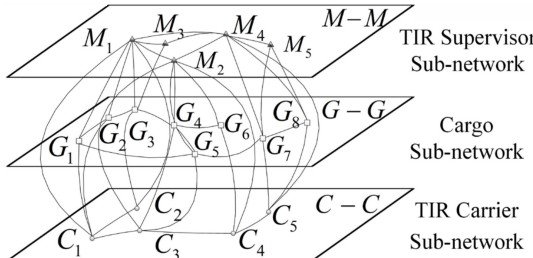

**Figure 1.** TIR transport structure diagram.

### 3.3. Structural Characteristics of TMTLSN

"Risk management" is a very important method in customs supervision, and "risk" is often considered by the customs as the possibility of criminals escaping customs supervision for tax evasion and smuggling. TIR-based multimodal transport may face greater hazards as a result of the mix of maritime and road transport modes. In $TMTLSN$, the main management method of the customs nodes in the $M - M$ layer sub-network is to obtain transport-related information through the TIR-EPD information management system and judge the risks before TIR transport starts. In this paper, we believe that these risks are going to cause disruption in $TMTLSN$. Customs at different ports check TIR carnets, vehicles, and seals, and there is no need to inspect if there is no risk.

Different customs ports can operate under different supervision procedures within the same nation; however, TIR ports of a certain country have relatively close cooperation with each other on a TIR transport route, while TIR ports of different countries operate differently. However, the capture probability will be increased after nodes jointly receive IRU training or exchange risk analysis methods internally. This also means that risks such as handling errors are more likely to spread among ports within a certain country. The customs will punish the carrier for the tariff loss and claim compensation through TIR carnet issuing bodies, guarantee agencies, and other supervisors; nodes in the $C - C$ layer sub-network will gradually establish a community structure with cluster characteristics, which is beneficial to industrial cluster and enterprise cooperation, after connecting with several enterprises to transport goods. This structure has the characteristics of tight connections between nodes within the community and loose connections between communities, as shown in Figure 2. Nodes in the $C - C$ layer sub-network can improve the business mode through cooperation or information exchange, but the risk in one node will also affect other nodes. The termination of TIR transport, for instance, will have an impact on other honest nodes transporting comparable goods if the risk brought on by the dishonest transport behavior of a node facing the temptation of interests is discovered by customs. That is to say, $M - M$ and $C - C$ layer sub-networks have the following characteristics: most nodes are not directly connected with each other, but most nodes can be connected to any other node in a few links. The network has connectivity, and the risk transfer speed is fast. Because of the nature of their small world network, a few connections can drastically alter network performance. At the same time, these two layers of sub-networks also have the nature of community structure; some risk may be generated in multiple nodes with a similar degree simultaneously, or the external risk may attack a node and spread through the connection or community.

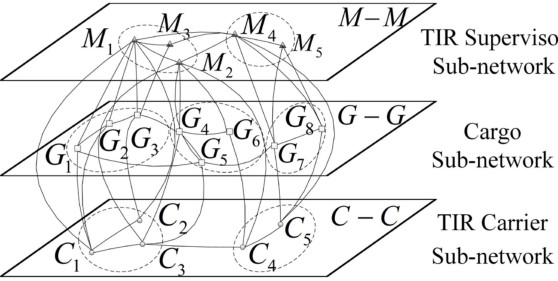

**Figure 2.** Community Structure in $TMTLSN$.

The cargo sub-network $G - G$ has the following characteristics: first, the non-uniformity of the structure refers to the uneven degree of nodes; the number of large-degree nodes is relatively small, and the number of medium or small-degree nodes is relatively large, because the certain kinds of goods that the TIR carrier is transporting or that are relevant to supervisors are few because of the high risk. Second, the growth of the structure refers to the continuous growth of the network scale due to the continuous emergence of new goods and new TIR carriers. Third, priority connectivity refers to the fact that the transport routes and loading modes of new TIR goods are generally based on the main goods with stable transportation volume, that is, they tend to connect large nodes. These characteristics lead to the $G - G$ layer sub-network having the property of a BA scale-free network. This network was proposed by Albert-László Barabási and Réka Albert. It has the following characteristics: (1) the scale of the network is constantly expanding; (2) new nodes are more likely to connect to nodes with higher connectivity. Additionally, because some goods are frequently transported simultaneously, this also forms a specific community structure and risk in the $G - G$ layer sub-network.

The connectivity within and between the three layers also has certain priority connectivity: the new transport enterprises joining the $C - C$ layer sub-network are more inclined to establish a cooperative relationship with the powerful large-degree transport enterprises, and new types of goods are more likely to be handed over to such enterprises for transport; the nodes in the $M - M$ layer sub-network are more likely to have learning and business communication with the large-degree nodes with high management levels; the degree node is more relevant to large-degree goods.

As shown in Figure 3, due to the structural nature of $TMTLSN$ and the pursuit of transport efficiency, the connection within and between each layer of the community is close or efficient. If $C_1$ generates the risk of dishonesty, the risk will spread rapidly within the community (as shown by the dotted arrow in Figure 3) and affect the goods in the $G - G$ layer sub-network through the connection. The failure of $TMTLSN$ occurs when the nodes in the $M - M$ layer sub-network break the connection by suspending TIR transport and carrying out further processing after identifying the risk. However, this does not ensure there is no operation failure of $TMTLSN$ if the nodes in the $M - M$ layer sub-network fail to deal with the risk because of management oversight or shielding of the nodes in the $C - C$ layer sub-network. On the contrary, temporary relaxation will lead to more frequent risks and, eventually, will cause the connection to break. The risk may also be generated by the $M - M$ layer sub-network first and then propagated back.

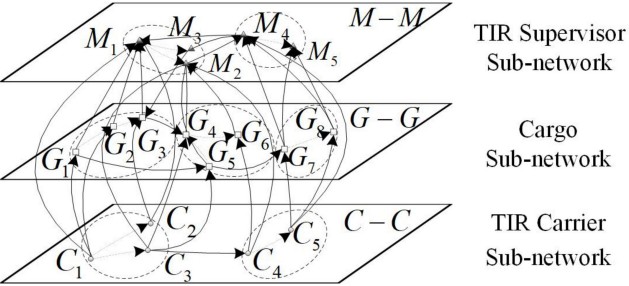

**Figure 3.** Risk communication diagram of $TMTLSN$.

### 3.4. Performance Parameters of TMTLSN

Hierarchy, matching, transmissibility and agglomeration are the four elements used to measure resilience [26]. Starting from these four attributes, this paper discusses the resilience of network structures one by one. This paper measures the performance of the super-network using four parameters to analyze the changes under risk [5].

Hierarchical intersectionality degree $C_\lambda$, which reflects hierarchy, is a measure index of connection properties that takes into account the difference of link attributes in the super-network. Because most of the nodes in three sub-networks of $TMTLSN$ are related to each other, in this paper, $C_\lambda$ represents the level of mutual connection between nodes

of the super-network at the same time. As the TIR association between node $i$ and $j$ is represented by a link, the number of links within the $M - M$ layer sub-network is $M_{ij}$, the number of links within the $G - G$ layer sub-network is $G_{ij}$, the number of links within the $C - C$ layer sub-network is $C_{ij}$, the number of links between the $M - M$ layer sub-network and $G - G$ layer sub-network is $MG_{ij}$, the number of links between the $G - G$ layer sub-network and $C - C$ layer sub-network is $GC_{ij}$, the number of links between the $M - M$ layer sub-network and $C - C$ layer sub-network is $MC_{ij}$, and $N$ is the total number of nodes in $TMTLSN$. The calculation formula of hierarchical intersectionality degree $C_\lambda$ in $TMTLSN$ is presented as follows:

$$C_\lambda = \frac{1}{N(N-1)} \sum_{i \neq j} \frac{MG_{ij} + GC_{ij} + MC_{ij}}{M_{ij} + G_{ij} + C_{ij} + MG_{ij} + GC_{ij} + MC_{ij}} \tag{1}$$

Proportionality coefficient $r$, which reflects the matching, is a measure index of the correlation and "attraction" between the degree and degree of network nodes. Large-degree nodes tend to connect with other large-degree nodes. The calculation formula is as follows:

$$r = \frac{e_t^{-1} \sum_m p_m q_m - \left[ e_t^{-1} \sum_m \frac{1}{2}(p_m + q_m) \right]^2}{e_t^{-1} \sum_m \frac{1}{2}(p_m^2 + q_m^2) - \left[ e_t^{-1} \sum_m \frac{1}{2}(p_m + q_m) \right]^2} \tag{2}$$

where $p_m$ and $q_m$ are respectively the degrees of two nodes $p$ and $q$, which make the $m$ link, and $e^t$ is the total number of links of the network.

Clustering coefficient $C$ reflects the agglomeration, indicating the degree of aggregation between nodes in the network, and represents the community characteristics in $TMTLSN$. The calculation formula is as follows:

$$C = \frac{1}{N} \sum_{i=1}^{N} \frac{2E_i}{k_i(k_i - 1)} \tag{3}$$

Network efficiency $L$ represents the connectivity efficiency, the degree of information exchange, and the smoothness of TIR routes between nodes in multimodal transport logistics, namely, the efficiency of multimodal transport logistics. The calculation formula is as follows:

$$L = \frac{1}{\frac{1}{2}N(N-1)} \sum_{i \neq j} \frac{1}{d_{ij}} \tag{4}$$

Among them, $d_{ij}$ is the shortest distance between two nodes $i$ and $j$ in the super-network, which is defined as the number of links between $i$ and $j$.

In $TMTLSN$, these four indicators will change in varying degrees under the attack of disruption caused by the risk addressed in Section 4.3.1.

The spatial centrality $S(i)$ reflects the transmissibility, indicating whether a node is associated with many other nodes in the overall space of the super-network, especially with many points with high centrality, which determines the degree of occupying a key position in the super-network and the transmission performance. The calculation formula is as follows:

$$S(i) = \sqrt{D^2(i) + B^2(i) + E^2(i)} \tag{5}$$

where $D(i)$ is degree centrality, $B(i)$ is betweenness centrality, $E(i)$ and is eigenvector centrality.

## 4. Empirical Analysis

### 4.1. TMTLSN Modeling Based on Empirical Analysis—Taking the Qingdao Port as an Example

The above-mentioned super-network model is designed to facilitate the study of simplified diagrams. To make the research more practical, a model that has all the structural features and properties of $TMTLSN$ based on the practical example should be established for simulation.

In the current TIR actual operation system, 77 countries in the world have become TIR members, namely TIR contracting parties, most of which are located in important regions along the "Silk Road Economic Belt", which integrates with the "Belt and Road Initiative" of China; thousands of ports have become TIR ports around the world; and over 34,000 transport operators have become TIR licensees, namely TIR goods carriers. The newly opened multimodal TIR-based transport routes have gradually extended from Central Asia and Russia to European countries. With the update of the routes, the farthest reach is extended to Sweden, the United Kingdom, Portugal, and other countries, and the goods transported by this kind of multimodal transport are mainly in the first 96 of the 98 chapters of goods classified by the Import and Export Commodities Classification; the actual TIR super-network will be large.

Considering the convenience of the study, this paper selects Qingdao Port, which is one of the most important multimodal transport ports in China, as the core transport node. Since 2022, Qingdao has seized the opportunity in the operation of Regional Comprehensive Economic Partnership to build a "Eurasian Express + TIR International Road Transport" hub, attracting a large number of goods from Japan and South Korea, such as precision instruments and semiconductor equipment. These goods are loaded in ports such as Osaka or Busan and arrive at Qingdao by sea. They may be trans-shipped in transportation hub Urumqi first, and then transported together with other goods there. Alternatively, they can also be directly transported directly from Qingdao to border ports such as Khorgos, Baktu, Alashankou, Irkeshtan, Kashgar, etc., and then transported to Central Asia via stable TIR transport routes such as China–Kazakhstan–Russia, China–Kazakhstan–Uzbekistan, or via neighboring Russia, and then cross Belarus and Poland to Europe. In recent years, through negotiations with Russia, Kazakhstan, Mongolia and other countries (on 1 December 2013, the Russian General Administration of Customs decided that the Far East and Siberia regions would not implement the TIR Convention; however, Russia's inland and other countries' ports were still TIR ports, and all the ports of China were negotiated by both China and Russia and signed the Agreement on International Road Transport between the Government of the People's Republic of China and the Government of the Russian Federation on 8 June 2018. The "open" here refers only to the opening of a port to TIR), the ports of Bishkek, Almaty, Mashatakov, Troysk, Boglaiqi, Blagoveshchensk, Khabarovsk, Birobidzhan, Yaalama, Ulan Bator and other ports have been accepted as ports on the TIR route. In addition to these ports, this study also includes ports such as Zinaz in Uzbekistan, Stolbutzi and Biavestok in Poland, Ansbach in Germany, and Alicante in Spain on the route through the European Union. The customs of these 30 ports in total are taken as nodes of the $M - M$ layer sub-network. According to customs statistics, export goods in the multimodal transport logistics between China and European are mainly concentrated in chapters 7, 8, 20, 39, 60, 64, 65, 69, 70, 81, 82, 83, 84, 85, 87, 95 and 96 (in this paper, the abstract induction of the goods nodes is carried out in accordance with the chapter in the Imports and Exports Commodities Classification. The General Administration of Customs prohibits the transport of HS codes from TIR lines into six types of alcohol and tobacco products, coded as 22.07.10, 22.08, 24.02.2.10, 24.02.20, 24.01 and 24.03.19, which are classified as goods under articles 22 and 24). According to the ranking, the goods with export value more than 50 million yuan in total are selected, involving 34 chapters, indicating 34 nodes in the $C - C$ layer sub-network. However, the export goods of each port in the TIR network are different, and the TIR carriers are also different, so the connection between different nodes of the $C - C$ layer sub-network and the other two layers is not the same. As a result of regional restrictions, a total of 55 TIR transport operators registered in Shandong and Xinjiang Custom District and TIR transit port are taken as a $C - C$ layer node. In this way, the super-network has 3 layers and 119 nodes, named $TMTLSN - QD$. The super-network model will be more complex as the TIR ports and transport routes increase.

### 4.2. Structure Characteristic Verification of TMTLSN in Qingdao Port

After building a super-network based on real figures and connections between nodes in Qingdao port, to conduct simulation research on the basis of the above super-network model, the complex network characteristics of each layer need to be verified and analyzed first in the simulation study.

First, we used MATLAB software to calculate the average link length and clustering coefficient of 10 ER random networks (a network proposed by and named after Paul Erdős and Alfréd Rényi; the points in the network are randomly distributed) with the same size as the $M - M$ layer sub-network and $C - C$ layer sub-network, and then compared the $M - M$ layer network and the $C - C$ layer network. The results are shown in Table 1 below.

**Table 1.** Comparison of $M - M$ Layer Network, $C - C$ Layer Network and ER Random Network Parameters.

|  |  | ER Random Network | $M{-}M$ Layer Network | $C{-}C$ Layer Network |
|---|---|---|---|---|
| Indicator | Average path length | 3.0146 | 2.6739 | 2.8701 |
|  | Average clustering coefficient | 0.0337 | 0.1867 | 0.1989 |

It was found that the average link length of the two networks is less than the average path length of the 10 random networks, and the clustering coefficient is greater than the average clustering coefficient of the 10 random networks. These two networks do not have isolated nodes in practice and have the nature of an NW small-world network (Watts and Strogatz proposed a WS small-world network model. In a small-world network, the connection status (degree) between the nodes is evenly distributed. Most nodes in the network are not adjacent to each other, but most nodes can be reached from any other point in a few steps. Newman and Watts proposed the NW small-world network model, replacing the "randomized reconnection" in the construction of the WS small-world network model with "randomized margining". The NW small-world network is named after them).

The degree distribution probability is the main indicator to measure whether a complex network has a scale-free property, and the degree distribution of the scale-free network follows the power-law distribution. The degree distribution data is fitted, and the results are shown in Figure 4. According to the fitting results, the power exponent $r = 2.47$ and $R^2 = 0.8527$, which is in line with the rule of the power exponential value of the scale-free network; thus, the $G - G$ network has a scale-free feature.

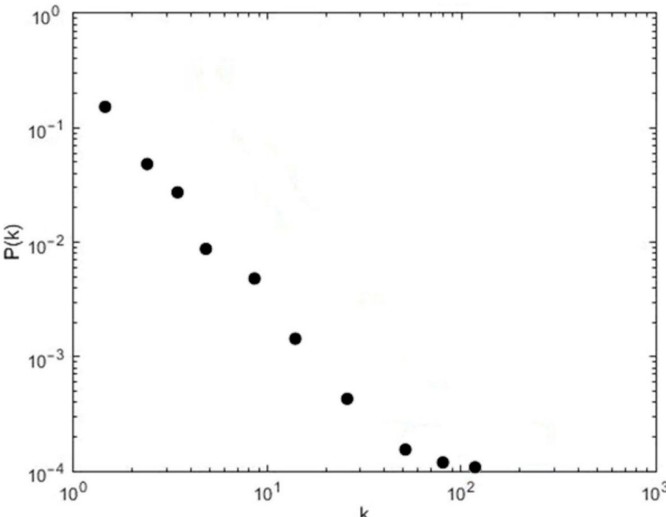

**Figure 4.** Log-log coordinate degree distribution fitting diagram of $G - G$ layer sub-network in $TMTLSN - QD$.

Therefore, $TMTLSN - QD$ performance analysis can be conducted based on super-network theory and methods. Using the NetLogo software, the 3D figure for $TMTLSN - QD$ is established, as shown in Figure 5 below:

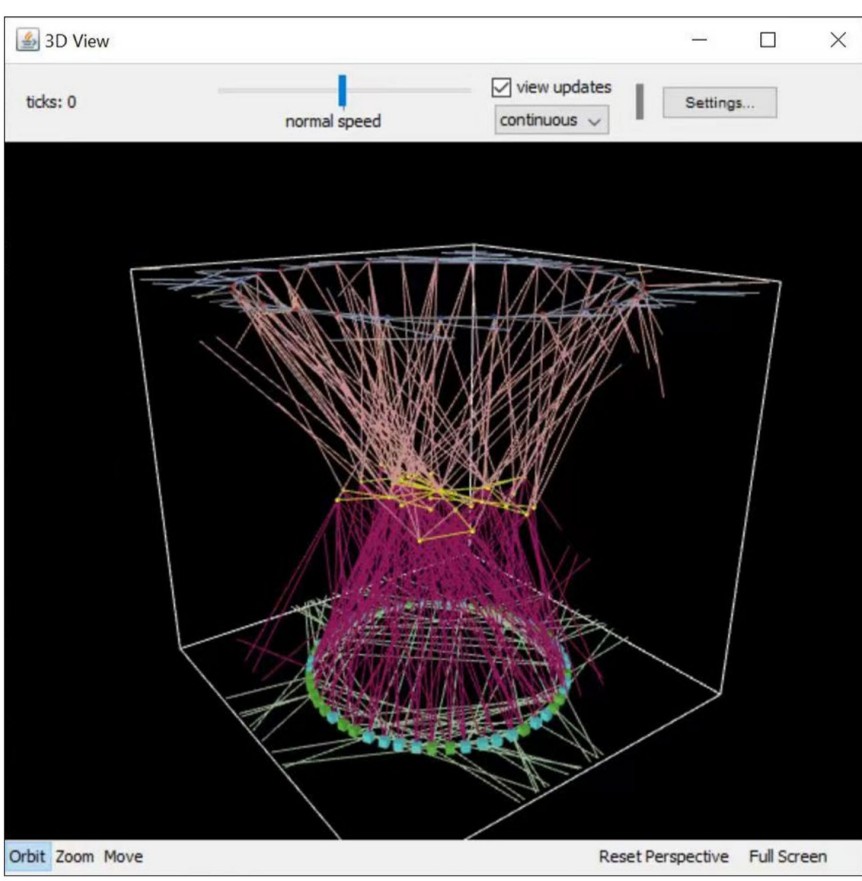

**Figure 5.** 3D model Structure of $TMTLSN - QD$.

### 4.3. Risks and Immunization Strategies in $TMTLSN$

#### 4.3.1. Risk in TMTLSN

Risk in the multimodal transport logistics super-network based on multimodal transport refers to various events or states that interfere with TIR transport and lead to the possibility of loss. This will not only harm individual nodes but also continuously diffuse through internode connections, causing interruption failure or even overall collapse in the multimodal transport logistics super-network (the overall collapse here refers to the interruption of transport of a licensee by a certain risk in TIR transport, interruption of transport routes or even TIR port closure in a certain area or country, such as the closure of TIR ports in the Far East region of Russia in 2013). To effectively identify the risk, this study collected typical violations and accident information of TIR transport worldwide through joint in-depth studies with the General Administration of Customs, the Ministry of Transport and the IRU (World Road Transport Organization, a TIR management agency authorized by the United Nations) and finally formed a TIR risk database containing 115 cases. As multimodal transport logistics is a collection and organic whole of multiple departments and institutions, such as goods carriers, customs, the IRU, the Ministry of Transport, transport, industry associations, etc., based on the database and combined with the characteristics of multimodal transport logistics and IRU's TIR transport risk database, 24 kinds of TIR transport risks are identified, as shown in Table 2. Any individual risk may be triggered by other risks or will trigger other risks. The occurrence of any disruption may lead to the interruption of multimodal transport logistics, and the relationship between the risks also reflects the link in the super-network to a certain extent.

**Table 2.** Possible risk of TIR-based multimodal transport logistics.

| Goods Related $R_1$ | | Related Administrative Organizations $R_2$ | | Related Goods Carrier $R_3$ | |
|---|---|---|---|---|---|
| Content | Proportion | Content | Proportion | Content | Proportion |
| Type of goods $R_{11}$ | 0.017391 | Mistakes in customs enforcement $R_{21}$ | 0.008696 | Credit risk of the enterprise $R_{31}$ | 0.026087 |
| Specific cargo nature $R_{12}$ | 0.008696 | Ineffective communication between industry associations and customs $R_{22}$ | 0.008696 | Subcontracting of goods $R_{32}$ | 0.06087 |
| Embargoed goods $R_{13}$ | 0.008696 | Ineffective communication between the Ministry of Transport and Customs $R_{23}$ | 0.017391 | Vehicle affiliation $R_{33}$ | 0.104348 |
| Damage to packaging of goods $R_{14}$ | 0.026087 | Mistakes in the management of the Ministry of Transport $R_{24}$ | 0.008696 | Vehicle reinforcement facilities damage $R_{34}$ | 0.104348 |
| Loss and damage of goods $R_{15}$ | 0.026087 | Unreasonable formulation of laws and regulations $R_{25}$ | 0.008696 | Low speed of transport $R_{35}$ | 0.069565 |
| False declaration or swap of goods $R_{16}$ | 0.026087 | Unreasonable route planning $R_{26}$ | 0.043478 | Secretly carrying of goods to smuggle $R_{36}$ | 0.113043 |
| Goods overweight $R_{17}$ | 0.043478 | Unreasonable operation in freight station $R_{27}$ | 0.008696 | Document forgery $R_{37}$ | 0.069565 |
| | | Inconsistent implementation of vehicle standards between different Customs $R_{28}$ | 0.017391 | Tax evasion $R_{38}$ | 0.104348 |
| | | | | False and concealed report $R_{39}$ | 0.069565 |

It should be noted that during the pandemic, some countries tested whether the goods carried the virus, and once such risks were detected, TIR would be stopped. However, in the post–COVID-19 era, such testing is no longer carried out for import, export and transit goods; thus, this article removes the risk types of goods carrying the virus when collecting information. Any one of these risks may be triggered or able to trigger other risks, and the occurrence of any risk has the potential to disrupt multimodal transport logistics. The correlation between risks also reflects the edge connection relationship in the super-network to some extent.

4.3.2. Risk Immunization Strategy

The small world characteristics of the sub-network make the risk spread rapidly; because the sub-network has scale-free characteristics, the impact of wave and scope caused by nodes with a large degree under the risk attack is much larger than other nodes. At the same time, the inter-layer link of the super-network also diffuses risk throughout the TIR-based multimodal transport logistics, bringing great challenges to transport safety and resilience. Therefore, in the operation process of multimodal transport logistics, it is often necessary to implement effective preventive measures and control strategies. These preventive measures and control strategies include targeted training or early warning for nodes in the multimodal transport logistics system according to risks, which is similar to vaccinations for people and patching of a computer program; this is the immune method in immune theory. In this paper, two methods of immunization are used to study the $TMTLSN - QD$ immune strategies: first, random immunization $(RI)$, which is the completely random extraction of some nodes in the network for immunization; second, the choice of immunization $(SI)$, which is the purposeful extraction of important nodes in the network for immunization. According to the spatial centrality index $S(i)$ of super-network nodes, the first 50% of nodes are selected for immunization.

## 5. Results

*5.1. Risk Simulation Results of TMLTSN*

Under the continuous attack of various risks, multimodal transport logistics may continue to produce link interruption faults, affecting the conduct of transport. Therefore, the simulation of this article adopts the opposite method of removing the connecting link with probability $DP$ to simulate the probability of various risks in Table 2 with $DP$ in $TMTLSN - QD$ of the Qingdao port, resulting in the connection interruption and failure. If the failure occurs, the connection will not be reconnected after being interrupted; in this case, the simulation is carried out on the basis of the model established by the Net Logo simulation software according to Formulas (1)–(4), and the change trend chart of the three risk-related properties of the super-network is calculated as in Figure 6.

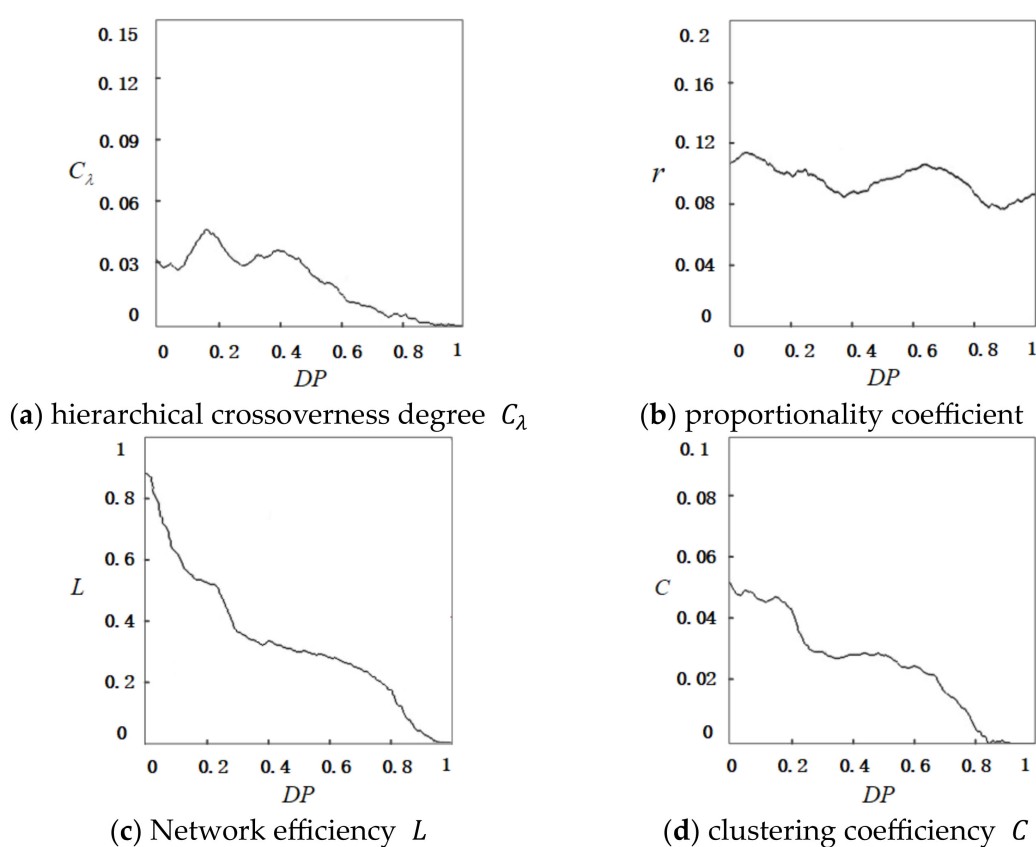

(**a**) hierarchical crossoverness degree  $C_\lambda$

(**b**) proportionality coefficient  $r$

(**c**) Network efficiency  $L$

(**d**) clustering coefficient  $C$

**Figure 6.** Four characteristic indicators of $TMTLSN - QD$ along with change diagram of $DP$.

The simulation results show that $C_\lambda$ has a peak value when the risk is about 20%, decreases slowly when the risk is between 20–90%, and falls to 0 when the risk increases to about 90%, indicating that the "huddling" and association among the three layers increase when the risk is small; these are gradually broken down under the risk impact, and the inter-layer contact fails and $TMTLSN - QD$ cannot run properly. The internal structure of each layer of $TMTLSN - QD$ is scattered, and the connection between layers is interrupted. In addition to the significant decrease when the risk increases to about 90%, $r$ generally maintains stable fluctuation, which indicates that the nodes in $TMTLSN - QD$ maintain the connectivity between nodes with a similar degree and are less affected by the risk. When the risk increases to about 30%, $L$ drops precipitously, which means that the connectivity efficiency between nodes drops rapidly to 0 in a short time when a large number of risks attack, and the transport in $TMTLSN - QD$ gradually stops. It should be noted that, in reality, the links between nodes in multimodal transport logistics can be reconnected. If goods are transported by multimodal transport other than sea and road transport, the connection efficiency will not be zero. Moreover, the connection efficiency being 0 does

not mean that the port is shut down or the international transportation of goods cannot be realized. Goods can still be transported through the port by the traditional temporary import–export mode before TIR, and customs will supervise the whole process according to the general trade method; however, the efficiency is far less than in TIR transportation. The time and cost required will be multiplied, and the value of $d_{ij}$ will be greatly increased. $C$ fluctuates slightly around 0 when the risk increases to about 80%, which indicates that when the risk is high, the community characteristics between nodes almost do not exist, the internal structure of each layer of $TMTLSN - QD$ is scattered, and the interlayer connection is also interrupted.

### 5.2. Resilience Simulation Results of TMLTSN

There are 118 nodes in $TMTLSN - QD$, and 50% of nodes are selected according to the two immunization strategies mentioned above to investigate the implementation effect under different immunization strategies. Assume that in each time step, the probability of infection risk of the immunized node is 30%, the node with infection risk can contact all of its neighbors, and the probability of recovering the connection between the original nodes after infection risk is 20%. The dynamic changes of network efficiency under the two immune strategies are obtained from the simulation model described above, as shown in Figure 7.

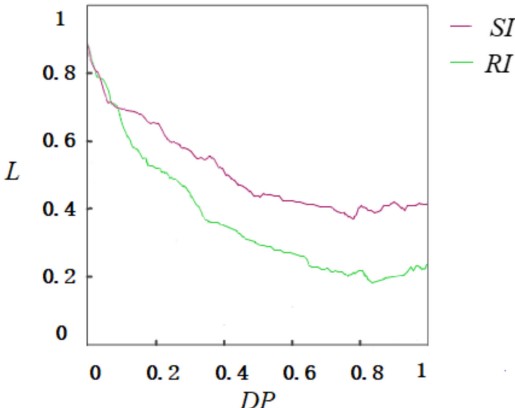

**Figure 7.** Dynamic change of network efficiency $L$ under two immune strategies.

Immunization of nodes is an effective means to prevent and control risk from spreading and causing damage in $TMTLSN - QD$; good immunization strategies should not only inhibit the spread of risk but also protect certain network structures and operations under the attack of risk.

It can be seen from Figure 7 that the results of the two immunization methods reflect the unbalanced phenomenon that the network efficiency decreases rapidly first and then slowly, because when the risk is small, the immunization of nodes causes the loss of some critical links in the super-network at the same time, resulting in the increase of the average length of the links between nodes; thus, the network efficiency decreases rapidly. With the increase of the risk, the decline of the super-network efficiency presents a trend of declining rapidly first and then slowly. This is because there are fewer connections between the remaining nodes in the later stage, and the change in connection efficiency is not as significant as in the initial stage. Some links between nodes are reconnected due to the immune strategy, so the curve finally shows a slight upward trend. This also indicates that the random immunization strategy is not as effective as the targeted selective immunization strategy in improving resilience.

## 6. Discussion

### 6.1. Key Findings

Multimodal logistics has become an important support for the global logistics system due to its significant advantages. In the post–COVID-19 era, global logistics will continue to face challenges, uncertainties and disruptions, and we need to further study how to achieve better multimodal transport organizational methods so that global logistics has a strong integrated and sustainable model to achieve sustained competitiveness and growth. This will increase resilience, avoid risks, and prevent disruptions, allowing the global economy and transport sector to recover from the impact of the pandemic as soon as possible. As mentioned in the literature review, existing research on multimodal transport has focused on issues such as transport mode combination, infrastructure impact and route selection, and risk management in the transport field has focused on supply chain risks. Theories and models related to super-networks help to address the management and decision-making of interconnections and impacts between multilevel networks; however, these have not been used for the resilience enhancement and risk management aspects of multimodal transport.

In this study, a model is established and simulated on the basis of both theory and practicality, and an empirical analysis is also carried out. The main findings are as follows:

1. TIR-based sea–road multimodal transport can better exploit the superiority of multimodal transport and is an important form of multimodal logistics in the post–COVID-19 era. However, the structure and performance of the logistics network formed by this multimodal transport differ from those of traditional multimodal transport due to the important role of customs and other regulatory agencies, cargo characteristics and enterprise carrier relationships. It is not sufficient to consider only the combination of transportation modes and transportation routes. In practice, it is especially necessary to consider how customs and other regulatory agencies adopt risk management to carry out effective supervision and enhance the resilience of the system.

2. The super-network model can well describe the complex relationship issues studied in this article, which involve the types of goods, transport companies, and government department management. Therefore, based on analyzing the characteristics of the TIR-based sea–road multimodal transport logistics network, a super-network can be established, which is composed of two NW small worlds and one BA scale-free network. Its characteristics can be described by the four performance parameters proposed in this article, and its resilience can be measured by simulating its changes under risk attacks.

3. In order to more effectively manage risks and enhance the system's resilience, customs and other regulatory agencies can employ immune strategies, and selective immunization is more targeted and effective than random immunization. Therefore, based on the establishment of the super-network, this study conducted simulation research on the risk-related characteristics and immune strategies brought about by the structure of the super-network. Overall, the simulation results in Figure 6 show a general trend of the four parameter values decreasing with increasing risks, and the simulation results in Figure 7 show that the performance can be enhanced by taking selective immune measures to resist risks and achieve the goal of resilience enhancement. Therefore, immunization should first focus on the nodes with higher spatial centrality in the super-network to avoid risks. These nodes tend to be the major consignees in TIR transport, having the highest spatial centrality intersecting with goods and management layers; immunizing them first will yield good results. Moreover, some important customs nodes, such as those located at the ports where the consignees are registered and operate TIR, should be immunized.

These views are consistent with the main idea of existing studies. However, as mentioned in the literature review, existing studies, in contrast to the simulation and empirical studies in this paper, are based on qualitative research; the basis for proposing views is not sufficient, and the suggestions given are not specific enough. This study is not only scientific but also provides some practical solutions.

*6.2. Practical Implications of the Study (Strategies, Challenges and Perspectives)*

In the TIR-based sea–road multimodal transport, resilience improvement and risk management are not two separate and independent aspects. They can be jointly addressed by seeking to work with authorities, academia, and enterprises to adopt some concrete and operable strategies. These strategies are further discussed in the following sections.

6.2.1. Strategies

1. Risk management strategies

(i) The "risk immunization and risk detection" strategy

Network efficiency $L$ is critical to the success of multimodal transport logistics as it governs the flow of time. According to the study in Section 5.2, an effective risk immunization strategy is essential to ensuring that certain $L$ values, namely the $TMTLSN$ connectivity, while preventing risks. In the context of TIR transport, regulatory agencies can take such measures as first delineating the scope of key carriers with large $C_\lambda$ value and then implementing marketing campaigns and training programs to enhance the enterprise's awareness of transportation integrity and risk prevention ability. Regulatory agencies should take the necessary prevention and control measures for the risks that these enterprises may be exposed to, strengthen the inspection of the main goods and vehicles that are prone to risks, and regularly carry out policy and business training for the carrier company and the drivers who carry out the transport, to improve the risk prevention and treatment control ability of each node in the whole transport system and thus ensure the safety of multimodal transport logistics. They should strengthen communication with enterprises and drivers during the immunization process to collect information and use the methods in this study to conduct risk detection and share risk information with other management agencies in a timely manner. The European Union set a good example during the COVID-19 pandemic by developing the "Galileo Green" APP, which provides real-time information on transit at member states' ports and alerts them of potential risks in a timely manner. It helps to prevent the occurrence of risks and ensures that customs and regulatory authorities are promptly informed when risks are detected.

(ii) Reduce structural non-uniformity strategy

$TMTLSN$ suffers from structural non-uniformity due to gaps in management between supervisor nodes, the characteristics of goods and carrier enterprises, and the uneven degree of nodes. Consequently, nodes with similar degrees may form links or experience the same risks simultaneously. To mitigate this risk, the TIR management agency may consider introducing some smaller-degree goods and micro-enterprises through management guidance and policy orientation while also increasing the overall level of management for all nodes to ensure a more even distribution of degrees throughout the $TMTLSN$. During the COVID-19 pandemic, there were cases of small enterprises transporting masks, protective clothing, and other pandemic prevention materials through TIR transport all over the world. These goods were rare in TIR transport in previous years.

(iii) Maintain moderate communal nature strategy

It can be seen from the $C$ simulation results that the nature of the community will diminish in the face of risk. To maintain cooperation within the community, regulatory agencies can organize exchanges and discussions, strictly enforce regulations and policies and disseminate information, and guide and regulate connections within and outside the community in a directional manner. By doing so, the nature of the community can be preserved, and risks can be resolved and jointly resisted through informal relationships within the community. For instance, in response to the COVID-19 pandemic, many countries are exploring ways to reduce close contact between people by leveraging TIR-based multimodal transport logistics. Customs authorities are encouraged to share pandemic prevention and control information to ensure the communal nature of the multimodal transport logistics network.

2. Resilience improvement strategies

(i) Set up "surplus" strategy

In order to allow recovery and response after $TMTLSN$ interruption, some "surplus" ("surplus" refers to the human or material resources that exceed the ordinary needs) configurations can be maintained in the $M - M$ layer sub-network with relatively poor functions, or some nodes can be trained to have other functions in addition to their specific functions, to deal with risks through elastic strategy mobilization. For example, data transmission in TIR can be outsourced, or data processing and analysis functions can be maintained in training, education, and scientific research institutions within the customs department. To promote the development of TIR-based multimodal transport, the government can introduce measures such as providing TIR vehicles with special channels and other conveniences. These measures can include encouraging TIR transport through port transferring and drop-and-pull mode, which can further improve the efficiency of TIR transport in customs clearance while ensuring safety. These special channels can also be utilized for other types of transport, thereby maximizing their benefits.

(ii) Control key nodes strategy

The $r$ simulation results suggest that customs and other regulatory bodies can effectively suppress risks in multimodal transport logistics by closely monitoring nodes and taking corrective action when risks are detected. Such action may involve carrying out special rectification and policy publicity for these nodes to prevent the spread of risks to similar nodes or those with possible connections. For example, when a sudden surge in the export volume of a certain type of goods is detected, it may indicate the risk of false trade. By controlling the target enterprise, similar enterprises can be identified, and the risk can be suppressed before it spreads. In the post–COVID-19 era, it is imperative for all countries to improve emergency response capacity by developing work mechanisms and emergency plans for strategic materials or emergency materials, based on the principle of centralized management and unified allocation. This includes equipping them with corresponding logistics guarantee channels to facilitate timely transportation of materials "point-to-point" to the designated locations. By implementing these measures, the security system in emergency situations can be improved, the capacity to respond effectively to future emergencies can be enhanced, and the anti-risk capability of multimodal transport logistics can be strengthened.

(iii) Manage weak connections strategy

The $C_\lambda$ simulation results indicate that in $TMTLSN$, both "strong connections" with high probability between nodes with similar degrees and "weak connections" (American scholar Granovetter's point of view is that the role of close ties among certain nodes in the network is sometimes not as close as those that are not closely linked) with low probability between dissimilar nodes should be ensured. Hidden risks in $TMTLSN$ often occur through "weak connection". For instance, enterprises with limited cooperation with other enterprises and a narrow business scope may declare a small amount of goods through the territory or collude with other enterprises in different customs areas to avoid management. Hence, these "weak connections" often become the entry point of risk management. To address this issue, regulatory agencies can introduce relevant transport support policies that encourage upstream and downstream enterprises in multimodal transport logistics, including production enterprises, trade enterprises, freight forwarding enterprises, and transport enterprises to participate in the multimodal transport among countries along the "the Belt and Road" initiative. They can provide various policies, funds and tax support for enterprises in the industry to alleviate the pressure of transportation and market development, enhance the enterprise's ability to resist risks, cultivate the backbone of TIR transport, and guarantee the supply chain system from the source.

### 6.2.2. Challenges and Perspectives

This study was applied in practice. In March 2023, a shipment of frozen beef departed from Ulaanbaatar, Mongolia, and transited through China to be shipped to Busan, South Korea. Due to the recent increase in the number of transit shipments of frozen meat, and to the fact that the transport enterprises encountered significant TIR operations at several ports, this shipment is considered the key node. Erlianhot Customs at the port of entry in China opened the container for inspection and found that eight boxes of goods were missing from the container while the seal was intact, which was later confirmed to be missed at the time of departure in Ulaanbaatar; the TIR operation was terminated and the product returned, avoiding loss for the consignee. This study helps customs to prevent risks by conducting inspections at the first port of entry when they detect the possibility of risks occurring.

However, not only do the "strong connections" between critical nodes need attention, but some "weak connections" also contribute to ensuring the connectivity of the multimodal transport logistics network. After the full reopening of TIR ports in China, TIR operations have not yet returned to pre-pandemic levels. Most of the TIR operations that have resumed are for the transportation of goods between Central Asian countries and China. Chinese customs and other regulatory agencies are working hard to collect the opinions of enterprises and promote participation in TIR-based combined sea–road transport by enterprises of all sizes, including small-scale ones. Through education and training institutions, more relatively niche routes are being established to promote the smooth operation of international multimodal transport logistics in the post–COVID-19 era.

It should be noted that there were previously two transport routes for cargo transport through China to Europe: (i) the China-Kazakhstan-Russia-Ukraine-Poland-Germany route to the Netherlands or neighboring countries, or via France to Spain; (ii) the China-Kazakhstan-Russia-Belarus-Poland-Germany route to the Netherlands or neighboring countries, or via France to Spain. Following the Russia–Ukraine conflict, the second route has become a more feasible option due to the difficulty of border crossing between Russia and Ukraine. Currently, restrictions of TIR have been relaxed at specific transit ports, that is, only the country of origin and destination are specified, but not the specific transit port. Transport enterprises can choose driving routes based on factors such as road conditions and adjust the route in a timely manner based on weather, road conditions, and vehicle conditions. Due to the impact of the Russian–Ukrainian conflict, Kuznica on the border between Belarus and Poland is temporarily as a result of refugee movement, and only Koroszczyn and Bobrowniki are operating normally; thus, transport enterprises often need to detour to the Kamenny Log border between Belarus and Lithuania, which may pose some unexpected risks to TIR. After the outbreak of the pandemic and the Russian–Ukrainian conflict, new opportunities emerged; for example, in November 2022, Russia decided to shift several roads, including the Transbaikalsk road crossing, into all-weather operation in order to increase the flow of goods between the two sides. In January 2023, the port resumed export processing of goods to Manchuria, China, and began transporting timber from Buryatia to Manchuria. Due to certain restrictions on the China–Europe Railway Express (Russian line) caused by the conflict, Kazakhstan seized this opportunity to upgrade the capacity of railway and transit TIR on its own territory and enhance its position as a trade corridor between China and Europe. These changes have brought challenges and more development opportunities to TIR-based Sea–Road Multimodal Transport Logistics, requiring a continuous search for better resilience enhancement and risk management methods.

### 7. Conclusions

In the post–COVID-19 era, countries are searching for better ways to improve their multimodal transport logistics. This paper conducts research from the perspective of TIR-based sea–road multimodal transport logistics. First, the concept of a TIR-based Multimodal

Transport Logistics Super-Network (*TMTLSN*) is defined, and a model is established using super-network theory and methods. Second, the relationship between nodes in a super-network and its system structure are analyzed, and its property indicators are proposed. Third, this study summarizes the possible risks that *TMTLSN* may encounter, conducts risk and resilience simulations, and conducts empirical analysis. The article not only has theoretical significance but also has practical value. With the current promising prospects and strong demand for multimodal transport logistics, the methods proposed in this paper can aid customs and other regulatory agencies with limited management resources to enhance their management methods, adopt better resilience improvement strategies, improve service levels, enhance multimodal transport logistics performance, safeguard international transport and global supply chain security, and promote the implementation of the Belt and Road Initiative.

However, this study is limited to sea–road multimodal transport based on TIR and cannot fully represent all operations of multimodal transport logistics. Improving the resilience and risk management of multimodal transport logistics in the post–COVID-19 era is a complex challenge in reality. In this study, the factors are simplified, such as the variety of goods in multimodal transport logistics and the different packing and stowage methods of goods, due to the limitations of mathematical modelling and computer simulation. This may have led to relatively idealized models and parameter settings that deviate from reality to some extent. In future research, the model structure of the super-network will be improved, the risk categories will be adjusted according to the actual situation, the algorithm and simulation will be optimized, and new forms of resilience and risk management improvement, such as anti-terrorism and intellectual property protection, will be studied. In addition, exchanges with customs and TIR administrations will be strengthened, research results will be better applied in TIR supervision, experience will continue to be summarized, and methods and applications will be optimized to achieve a better system. More importantly, the scope of research should be broadened to a wider range of multimodal logistics resilience improvement and risk management strategies. This challenging research topic has practical significance and value and will be further studied in follow-up research.

**Author Contributions:** Methodology, R.L.; Software, R.L.; Formal analysis, R.L.; Investigation, Y.Y.; Resources, W.L.; Data curation, Y.Y.; Writing—original draft, R.L.; Writing—review & editing, W.L.; Funding acquisition, R.L. All authors have read and agreed to the published version of the manuscript.

**Funding:** This research was funded by the National Social Science Fund of China (Grant No. 21BGL218).

**Data Availability Statement:** Not applicable.

**Conflicts of Interest:** The authors declare no conflict of interest.

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
