# Peer review of "Resilience Improvement and Risk Management of Multimodal Transport Logistics in the Post–COVID-19 Era: The Case of TIR-Based Sea–Road Multimodal Transport Logistics"

_sustainability, doi:10.3390/su15076041_

Round 1
Reviewer 1 Report
The submitted paper is devoted to important risk management issues and resilience improvement for multimodal transportation. The novelty of provided study methodic and obtained results are impresses.
But the following suggestions were made after the paper reading:
1) English can be improved. It will be good to recheck the manuscript on grammar and language style, starting from abstract;
2) Some keywords fully duplicate the paper title. I recommend changing these phrases to another in keywords. This help to make Internet searches wider.
3) The issue of novelty is well highlighted in the introduction, but I did not find any citation of literature in this part. This fact is very strange because a submitted study is not the first in this direction of research. The opinion of previous scientists in this sphere has additionally justified the significance of presented thoughts in this part.
4) In the literature review, please change the type of references in parentheses. They should not be in uppercase.;
5) Problems with formula design throughout the content. Please, use guidelines for authors;
6) Line 362, Unfortunately, I do not know the route “China-Kazakhstan-Ukraine”. First, this route is not correct because Ukraine doesn’t have a joint border with Kazakhstan. Please present the correct option of this cargo multimodal transportation or additionally explain what was meant. This may be presented transit variant. Second, how can it now use this route during the war period? I recommend explaining the actuality of this supply chain.;
7) Please add part Discussions. This help to better understand of obtained results for future readers.
8) Please, see point 6 of my suggestions. I want to return to these recommendations because the achieved results must correspond to the current time. I understand, that study was carried out before the war and sanction period and these aspects did not account for in the submitted manuscript. But for highlighting actuality, maybe it will be good to add some part about updating obtained results according to current situations at regions of recipient countries. Also, I recommend adding a part about how the proposed models will be work in Post-War-Period because we will see many new factors which will be born (create) new risk types. Please, think about this point's recommendations.
Author Response
We gratefully acknowledge the editor and the anonymous reviewers for the valuable suggestions and comments. After seriously considering the suggestions and comments, we prepared a revised version of the manuscript. The corresponding responses are listed as follows.

Reviewer 2 Report
The article establishes the super-network model of TIR-based multimodal transport logistics, analyzes its system structure, and carries out simulation research on resilience improvement and risk management. The article is relevant and well written, it has the potential to be published in Sustainability. The improvement suggestions are as follows.
General comments:
- The introduction should include the structure of the paper.
- Please expand the literature of the paper. What is the source of the following data: “TIR has greatly simplified the Customs procedures, shortened the Customs clearance time and has saved up 79 to 58% of transportation time and 38% of transportation cost”. What is the source of this list: “Hierarchy, matching, transmissibility and agglomeration are the four elements to measure resilience.” Furthermore, when were the TIR ports were fully reopened?
- Conclusions should highlight what this paper brings as added value to what we already know in current international literature. Conclusions should also present the international implications of the results, and could briefly reflect on recent effects of the Russian-Ukraine war on the China-Europe logistics corridor.
- Does sharing and exchanging supervision methods facilitate tax evasion and smuggling for those who have access to insider information on these methods?
Specific comments:
- A full expansion of an abbreviated word or term should be provided at the first use of each abbreviation. (i.e. NW in line 396, EPD in line 193, BA in line 276) Please review all abbreviations.
- Please correct Table 2 that has Chinese characters.
- Section 4 title is too long and should be revised.
- Incorrect reference to 1.3 in line 339 and 416.
- There are two formula (1) in the article, line 210 and line 319. Please consider removing the first, it may be unnecessary. The structure diagram is a good representation of the super network.
- Please do not refer to a single author (line 419).
- The reviewer suggests to extend future research to risk detection, as one strategy to prevent risk occurrence.
Author Response

(The authors gave the same response as above.)

Reviewer 3 Report
The current manuscript tries to improve resilience and risk management of Multimodal transport logistics after COVID-19 pandemic and considered TIR (Transports Internationaux Routiers) sea-road multimodal transport logistics as the case study.
The methodology of the article may contain some novel theoretical notions but the case study and conclusion need serious revision.
1. The last paragraph of the introduction should contain the structure of next sections of the article.
2. Some of the abbreviations are not clarified in the manuscripts like TIR and ER random network.
3. Table 2 contains some illustration in Chinese language!
4. Extracting of the data is not explain clearly in section 4.2. Are they exactly according to the case which considered in the title of the article?
5. The authors should explain that “How the extracted data are related to after or before COVID-19?” Moreover, they should show the improvement which achieved after applying their proposed approach.
6. Is it possible that equation (5) and its related definitions transfer to section 3?
7. Conclusion is too short! I think the reason is the lack of transparency of the data used in the case study. The authors should have a brief review about the previous sections in section 6 and then discuss about the advantages and disadvantages of the proposed method based on the case study achievements.
Author Response

(The authors gave the same response as above.)

Reviewer 4 Report
The mentioned contribution is prepared at a high-quality level. It contains several interesting and original professional and scientific outputs. The theoretical and practical benefits are considerable and significant.
However, I recommend to expand the conclusion with specific knowledge so that the readers understand whether and how the main goal of the paper has been met. It is also possible to add a discussion in the conclusion, where the mentioned elements could be supplemented.
It would also be suitable to add at least 3-5 more references that relate to the issue of transport and logistics in the context of the COVID-19 pandemic. These are several scientific contributions that were addressed within the specific scientific research activity at the University of Žilina in Slovakia.
I also recommend to check your English and to correct potential mistakes.
Author Response

(The authors gave the same response as above.)

Round 2
Reviewer 3 Report
The authors has been response to all of the given comments properly.